# Effect of patient and treatment factors on persistence with antihypertensive treatment: A population-based study

**Sara Malo** [1,2]*, **Isabel Aguilar-Palacio** [1,2], **Cristina Feja** [2,3], **María Jesús Lallana** [2,4], **Javier Armesto** [4], **María José Rabanaque** [1,2]

**1** Department of Preventive Medicine and Public Health, University of Zaragoza, Fundación Instituto de Investigación Sanitaria de Aragón (IIS Aragón), Zaragoza, Spain, **2** Grupo de Investigación en Servicios Sanitarios de Aragón (GRISSA), Zaragoza, Spain, **3** Directorate of Public Health, Government of Aragon, Zaragoza, Spain, **4** Aragon Healthcare Service, Zaragoza, Spain

* smalo@unizar.es

**Data Availability Statement:** Data were provided by the regional Government of Aragon, Spain, so authors do not have permission to share the data. The permission obtained from the Aragon Health

## Abstract

### Purpose

To analyze patterns of antihypertensive drug use among new users in a Southern European population, and identify patient- and treatment-related factors that influence persistence.

### Methods

This is a retrospective observational study of new antihypertensive drug users aged ≥40 years in Aragón, Spain. Information on antihypertensive drugs (2014–2016) prescribed and dispensed at pharmacies via the public health system were collected from a regional electronic population-based pharmacy database. Persistence was assessed using the gap method. Kaplan-Meier and Cox regression analyses were conducted to analyze patterns of use and factors that influence persistence.

### Results

The 25,582 new antihypertensive drug users in Aragón during the study period were prescribed antihypertensive drugs in monotherapy (73.3%), fixed combination (13.9%), free combination (9.1%), or other (3.7%). One in five received antihypertensive drugs within 15 days of the prescription date, but not after. During the first year of follow-up, 38.6% of the study population remained persistent. The likelihood of treatment discontinuation was higher for participants who were male, aged ≥80 years, and received an antihypertensive drug in monotherapy compared with fixed combination.

### Conclusion

Overall persistence with antihypertensive therapy was poor, and was influenced by the sex, age and type of therapy. Fixed combinations appear to be a good choice for initial therapy, especially in patients with a higher risk of discontinuation. Nonetheless, adverse drug effects and the patient's preferences and clinical profile should be taken into account.

Sciences Institute (IACS) imply the exclusive use of the data by researchers who authored the present study. Thus, this information cannot be published or shared with other institutions. Data access requests should be addressed to the IACS through https://www.iacs.es/.

**Funding:** This work was supported by the Grupo de Investigación en Servicios Sanitarios de Aragón (GRISSA) [B09-17R] of the IIS Aragón, and funded by the regional Government of Aragón, Spain and the European Fund for Regional Development (FEDER). The funders had no role in study design, data collection and analysis, decision to publish, or preparation of the manuscript.

**Competing interests:** The authors have declared that no competing interests exist.

## Introduction

Antihypertensive drugs have proven to significantly reduce cardiovascular events by lowering blood pressure [1]. However, health outcomes in patients treated with these drugs do not always correspond with expectations [2].

The European Ascertaining Barriers to Compliance (ABC) project developed a taxonomy to standardize both the medication-taking behavior terminology and the evaluation of medication adherence [3]. According to this taxonomy, adherence is the process by which patients take their medications as prescribed, and has three different components or phases: initiation, implementation and discontinuation. Persistence with therapy refers to the last phase, and it is defined as the time between initiation of therapy and the last dose, which immediately precedes discontinuation.

Poor treatment persistence is the primary cause of loss of effectiveness of preventive cardiovascular medication [4, 5], and contributes to increases in healthcare utilization and overall healthcare costs [6].

The literature describes antihypertensive drug persistence rates ranging from 35% to 92% after 1 year [7]. However, accurate comparison of published results is hindered by differences in patient populations, follow-up duration, definitions of persistence, and data sources.

The use of pharmacy records as sources of information for epidemiological drug utilization studies has recently become widespread [8]. These sources can be used to investigate relationships between patterns of drug use and population and treatment characteristics. In the case of antihypertensive drugs, assessment of persistence using this kind of data poses some methodological challenges, mainly owing to the variety of potential drugs, doses, drug combinations, and indications. Moreover, most of the published studies classify antihypertensive drug users whose regimens are switched or added to during the follow-up as discontinuers, providing an unreal estimate of persistence.

The analysis of factors that influence persistence is of particular interest for the management of cardiovascular disease, and there is some controversy in the literature regarding the influence of the sex, age, initial drug class and type of therapy on persistence to antihypertensive drug treatment [9–12].

In this study, we describe patterns of antihypertensive drug use among new users in a Spanish population and analyze the influence of patient demographic characteristics and treatment-related factors on persistence.

## Material and methods

### Study design and data source

This retrospective observational study of antihypertensive drug users was conducted in the region of Aragón in northeastern Spain. Data were collected from an electronic population-based pharmacy database; the Information System for Medication Consumption in Aragon. It collects the following information for all prescriptions dispensed in pharmacies in Aragon via the public National Health Service: anonymous patient code, patient sex and birth date, dispensing date, Anatomic Therapeutic Chemical (ATC) code of the prescribed drug, number of Defined Daily Doses (DDD), and number of packages dispensed.

This source is complete, contains validated data, and has been used in previous drug utilization studies [13].

Drugs were identified according to the ATC/DDD classification [14], World Health Organization 2018. We included all therapeutic subgroups indicated for arterial hypertension treatment (ATC codes C02 [antihypertensives], C03 [diuretics], C07 [beta-blocking agents], C08 [calcium channel blockers], and C09 [agents acting on the renin-angiotensin system]).

## Study population

The study population consisted of all individuals aged ≥40 years in the region of Aragón who began antihypertensive treatment for the first time (new users) during 2015. Individuals <40 years were not considered because of the low frequency of hypertension at this age.

The population in Aragón on January 1st 2015 was about 1.3 million, of which about 752,000 people were aged ≥40 years and 99% were covered by the public healthcare system.

## Outcome variables

New users were defined as those that had not been prescribed any antihypertensive drug during the 365 days preceding the index date (i.e., the date of the first prescription). For each subject, age on the index date was calculated.

New users were stratified into four cohorts based on the type of therapy received during a 15-day period after the index date (index period). The consideration of the index period was because antihypertensive therapy prescribing is initiated in a gradual manner in certain patients, depending on their needs and response. Taking into account all drugs prescribed during this first period may therefore reduce the risk of misclassification of antihypertensive drug users:

a. **Monotherapy:** individuals treated with a single active principle during the index period.

b. **Fixed combination**: individuals treated with multiple active principles combined in a single formulation during the index period.

c. **Free combination**: individuals treated with a combination of single active principles in different formulations during the index period.

d. **Multiple mix**: individuals that received a treatment other than the aforementioned therapies (e.g. monotherapy and a fixed combination) during the index period.

Persistence was defined as the continuation of treatment from the index date to the date of discontinuation, and was evaluated using the gap method. Antihypertensive drug users were considered non-persistent or discontinuers if, during the follow-up period, the database revealed a gap more than twice the duration of the preceding prescription during which no medication was dispensed. This gap length was based on previous studies [15, 16]. The number of days of available medication was estimated based on the number of DDD in the most recently dispensed prescription. Therefore, the maximum gap allowed was estimated as twice the number of DDD. Subjects were censored if no new prescription was dispensed during course of the permitted gap, or upon reaching the end of the study period. Thus, the year 2014 corresponded to the "wash-out" period, 2015 to the study "recruitment" period, and 2016 to the "follow-up" period.

Antihypertensive drug users were classified according to their pattern of use as *persistent*; *discontinuers*; *discontinuers who restarted* during the study period; or *spot users*. Spot users were those that filled prescriptions exclusively during the index period, i.e. within 15 days of the index date. Upon identification, spot users were excluded from subsequent analyses.

In both the monotherapy and fixed combination cohorts, the initially prescribed therapy could have been changed during the follow-up period. *Switching* was defined as a switch from a single active principle to a different active principle or from a fixed combination to a different fixed combination or a free combination. *Add-on* was defined as the addition of a separate drug to the single active principle or the fixed combination initially prescribed. A change from a single active principle to a fixed combination in which one of the constituent drugs belonged to the same therapeutic subgroup was also considered *add-on*.

On the other hand, the type of cohort influenced the type of persistence assessed, i.e., class persistence (persistence with a medication from the same antihypertensive drug class) or therapy persistence (persistence with any antihypertensive medication). In line with previously published methodologies [8, 10], we measured class persistence in the monotherapy and fixed combination cohorts, and therapy persistence in the free combination cohort.

New users of ATC code C02 drugs and those assigned to the multiple mix cohort were not included in the persistence nor in subsequent analyses because of their low frequency and heterogeneity.

## Statistical analysis

The sex and age profile of the study population and the most prescribed drugs in each of the study cohorts were described. Persistence was analyzed using the Kaplan-Meier method, and differences between user groups using the log-rank test. Median time to discontinuation was estimated and Cox regression analyses were performed to identify factors influencing the probability of discontinuation, considering the following independent variables: sex, age group and type of therapy.

A sensitivity analysis was performed to account for the possible influence of the selected gap length and the number of DDD as a measure of the daily dose on participant classification. The proportion of persistent users and the median time to discontinuation were recalculated for each cohort by applying fixed gap lengths of 30, 60 and 90 days, based on previous research [9, 16, 17].

All analyses were performed using Stata version 12.1 (StataCorp, College Station, TX, USA).

## Ethical information

The research protocol of the study was approved by the Aragon Health Sciences Institute (IACS) and the Aragon Health Service, who provided us the corresponding pharmacy claims data. All data used were anonymized and the results aggregated, making patient identification impossible. Neither consent to participate nor additional ethics approval were therefore needed.

## Results

### Characteristics of new antihypertensive drug users and prescribed treatments

In total, 25,582 new antihypertensive drug users aged ≥40 were identified. The majority were assigned to the monotherapy cohort (73.3%), followed by the fixed combination (13.9%), the free combination (9.1%) and the multiple mix (3.7%) cohorts (Table 1). The proportion of subjects receiving each type of therapy differed according to sex and age (p<0.001). The frequency of prescription of antihypertensive drugs in free combinations as initial treatment increased with age, from 7.3% in the 40–59 years' age group to 13.5% in those aged ≥80 years.

The regimens most commonly prescribed in monotherapy or in fixed combination during the index period are shown in Fig 1. The most widely used regimen was angiotensin-converting-enzyme inhibitors in monotherapy (31.3%), followed by diuretics (21.9%), angiotensin receptor blockers (13.2%), and beta blockers (12.8%). On the contrary, the most used drugs in the free combination cohort (Fig 2) were diuretics (representing 32.7% of the total number of drugs prescribed to this cohort during the index period), followed by angiotensin-converting-enzyme inhibitors (23.0%) and beta blockers (20.0%).

**Table 1. Distribution of new antihypertensive drug users aged ≥40 years in Aragón (Spain) by sex, age group, and type of therapy during the first 15 days of treatment.**

| | Sex | | Age group | | | |
|---|---|---|---|---|---|---|
| | Women n (%) | Men n (%) | 40–59 years n (%) | 60–79 years n (%) | ≥80 years n (%) | Total n (%) |
| Monotherapy | 8,840 (72.0%) | 9,907 (74.5%) | 7,659 (75.1%) | 7,814 (72.4%) | 3,274 (71.2%) | **18,747 (73.3%)** |
| Fixed combination | 1,693 (13.8%) | 1,857 (14.0%) | 1,435 (14.1%) | 1,571 (14.6%) | 545 (11.9%) | **3,551 (13.9%)** |
| Free combination | 1,247 (10.2%) | 1,097 (8.3%) | 747 (7.3%) | 978 (9.1%) | 619 (13.5%) | **2,344 (9.1%)** |
| Multiple mix | 496 (4.0%) | 444 (3.3%) | 353 (3.5%) | 427 (4.0%) | 160 (3.5%) | **940 (3.7%)** |
| Total | **12,276 (100.0%)** | **13,305 (100.0%)** | **10,194 (100.0%)** | **10,790 (100.0%)** | **4,598 (100.0%)** | **25,582 (100.0%)** |

Abbreviations: n, number; %, percentage.

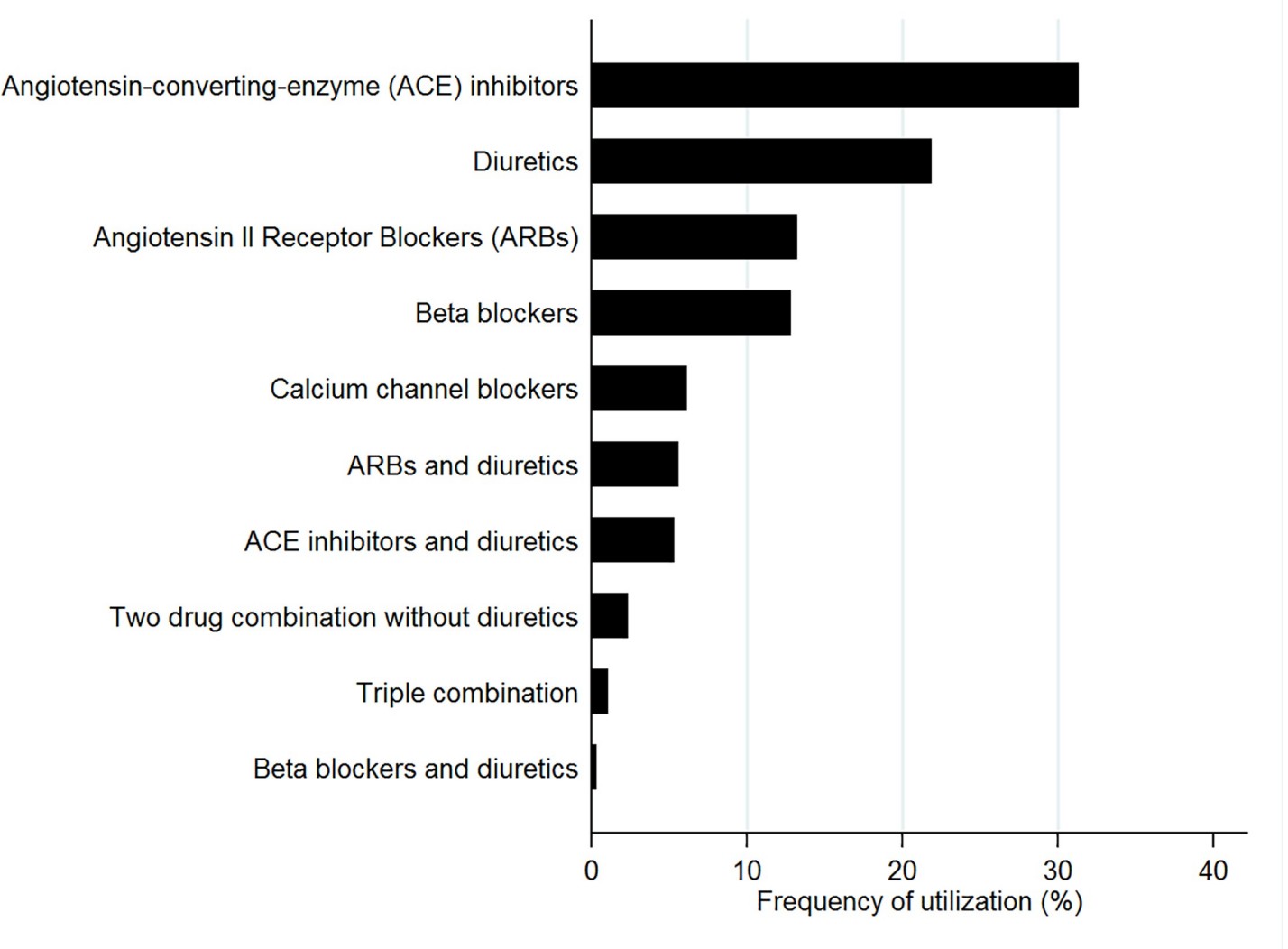

**Fig 1. Frequencies of the antihypertensive regimens most commonly prescribed as monotherapy or fixed combination to new users in Aragón (Spain).** The values indicated represent the percentage of new antihypertensive drug users treated with each antihypertensive regimen either as monotherapy or fixed combination during the first 15 days of therapy relative to the total number of new users.

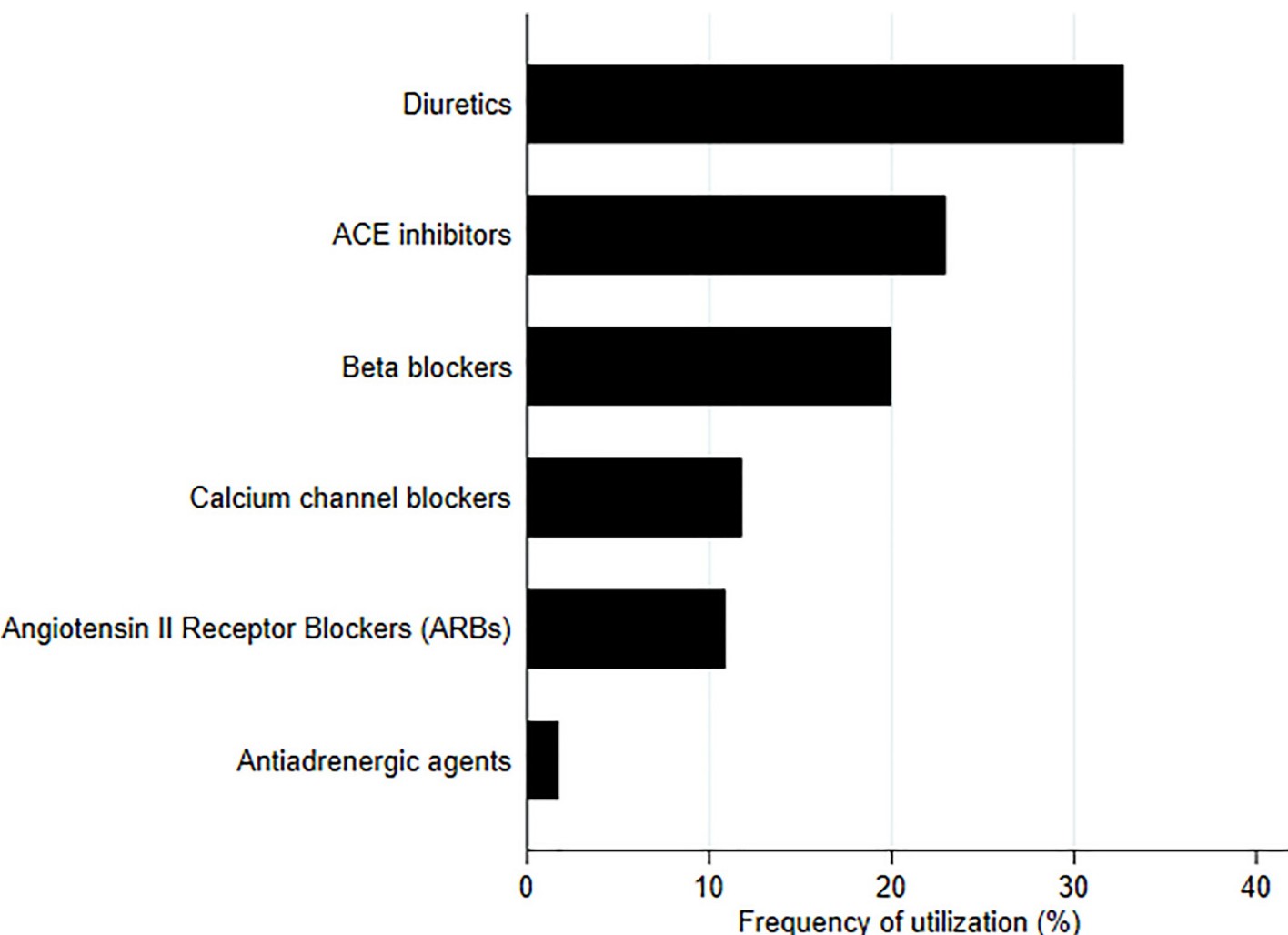

**Fig 2. Frequencies of the therapeutic subgroups most commonly prescribed as free combination to new users in Aragón (Spain).** The values indicated represent the percentage of prescriptions of each therapeutic subgroup relative to the total number of prescriptions prescribed as free combination to new antihypertensive drug users during the first 15 days of therapy.

### Pattern of use according to sex, age group and initial type of therapy

Comparison of persistence rates by sex (Fig 3) and by age group (Fig 4) revealed significant differences (log-rank, p<0.001). Women were persistent for longest compared with men, while those aged ≥80 discontinued therapy much earlier than the other two age groups. In particular, 25.8% of users aged ≥80 remained persistent for 1 year from the index date, followed by users aged 60–79 (40.6%) and users aged 40–59 (42.1%) (Table 2).

The proportion of persistent users also varied depending on the type of therapy prescribed (Table 2). Overall, 5,771 users (38.6%) starting antihypertensive therapy as monotherapy were classified as persistent. In the fixed combination cohort, there were 1,263 (43.3%) persistent users. The free combination cohort had the lowest proportion of persistent users (32.3%), as well as the greatest proportion of discontinuers who subsequently restarted treatment (50.6%). Analyses by age group and type of therapy confirmed that users aged ≥80 showed a lower proportion of persistent users independently of the type of therapy (p<0.001). There were not significant differences in the proportion of persistent users by type of therapy among the 40–59

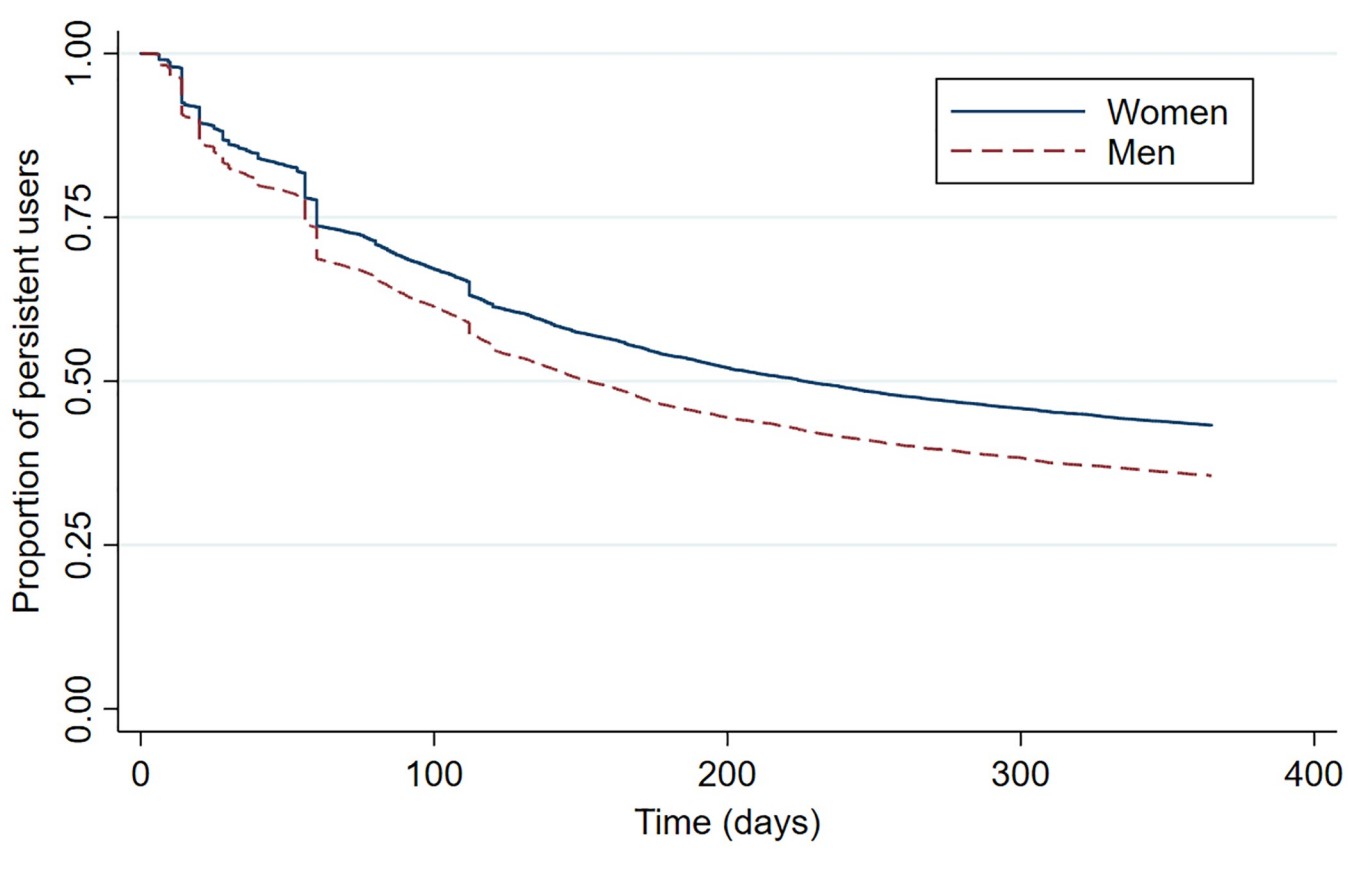

**Fig 3. Antihypertensive drug persistence during the first year of treatment in users from Aragón (Spain) according to their sex.**

and the 60–79 age groups, except among the free combination users (39.2% versus 32.8%, p = 0.006).

Of the total new antihypertensive drug users analyzed, 5,005 (19.6%) were spot users. By type of therapy, distribution of spot users was as follows: 20.0% in the monotherapy cohort, 17.7% in the fixed combination cohort and 13.7% in the free combination group.

Among the persistent users in the monotherapy cohort, 765 (13.3%) switched to another drug during the follow-up period, and 902 (15.6%) were prescribed an add-on medication.

Among the persistent users in the fixed combination cohort, 252 (20.0%) switched medication and 96 (7.6%) were prescribed an add-on medication.

Differences in the median time to discontinuation were observed depending on the type of therapy received: monotherapy cohort, 171 days; fixed combination cohort, 248 days; free combination cohort, 148 days.

## Multivariate analysis results

Table 3 shows the results obtained in both the bivariate and the multivariate analyses. Men were more likely to discontinue antihypertensive therapy than women (HR, 1.20; 95%CI, 1.15–1.24). Compared with the 40–59 years' group, the likelihood of discontinuing therapy was significantly higher for the oldest age group (≥80 years) (adjusted hazard ratio [HR], 1.45; 95% confidence interval [CI], 1.38–1.53). As regards the type of therapy, individuals prescribed

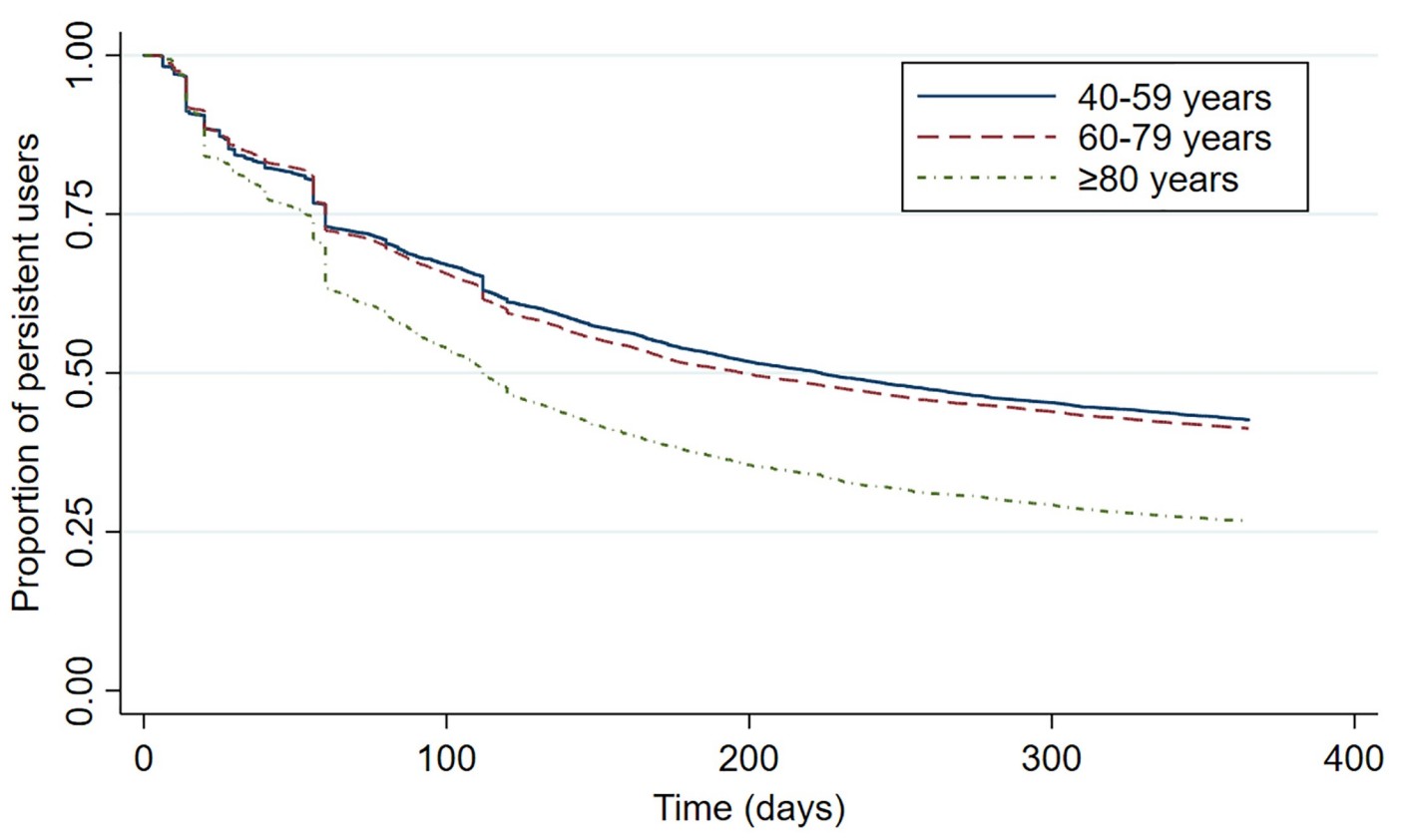

**Fig 4. Antihypertensive drug persistence during the first year of treatment in users from Aragón (Spain) according to their age group.**

a fixed combination had in both the univariate and the multivariate analyses a lower likelihood of discontinuation than those receiving monotherapy. Conversely, the association between prescribing of a free combination and treatment discontinuation did not reach statistical significance.

In the sensitivity analyses (S1 Table), the proportions of new antihypertensive users considered treatment persistent were 1,7%, 42.3% and 59.0% applying gaps of 30, 60 and 90 days, respectively. These results should be compared with the 38.6% of persistent users obtained when applying a gap of 2xDDD. Median time to discontinuation was of 46 and 228 days with gaps of 30 and 60 days, respectively. For the gap of 90 days and also for the gap of 60 days in the case of the free combination cohort, median time could not be calculated since less than 50% of users had discontinued 365 days after initializing treatment. The proportion of persistent users obtained with the gap of 60 days was similar to the obtained with the gap of 2xDDD for all the types of therapy () except for the free combination cohort, where 51.4% of users were classified as persistent, instead of 32.3%.

## Discussion

The findings presented here help further our understanding as to the factors that influence the initial prescribing of antihypertensive drugs and their subsequent use.

In our study population, less than half (38.6%) of the new antihypertensive drug users remained persistent after 1 year of treatment. Evidence suggests that measurement of

**Table 2. Classification of new antihypertensive drug users aged ≥40 years in Aragón (Spain) according to the age group, the initially prescribed type of therapy and the pattern of use over 1 year from the index date.**

| Type of therapy | Discontinuers n (%) | Discontinuers who restart n (%) | Persistent n (%) | Total n (%) |
|---|---|---|---|---|
| *Age group 40–59 years* | | | | |
| **Monotherapy** | 1,098 (18.2%) | 2,424 (40.1%) | 2,527 (41.8%) | **6,049 (100.0%)** |
| **Fixed combination** | 246 (19.9%) | 428 (33.7%) | 564 (45.6%) | **1,238 (100.0%)** |
| **Free combination** | 84 (12.4%) | 327 (48.4%) | 265 (39.2%) | **676 (100.0%)** |
| **Total** | 1,428 (17.9%) | 3,179 (39.9%) | 3,356 (42.1%) | **7,963 (100.0%)** |
| *Age group 60–79 years* | | | | |
| **Monotherapy** | 1,126 (17.6%) | 2,644 (41.4%) | 2,614 (41.0%) | **6,384 (100.0%)** |
| **Fixed combination** | 304 (23.8%) | 415 (32.5%) | 560 (43.8%) | **1,279 (100.0%)** |
| **Free combination** | 137 (17.1%) | 403 (50.2%) | 263 (32.8%) | **803 (100.0%)** |
| **Total** | 1,567 (18.5%) | 3,462 (40.9%) | 3,437 (40.6%) | **8,466 (100.0%)** |
| *Age group ≥80 years* | | | | |
| **Monotherapy** | 628 (25.0%) | 1,257 (50.0%) | 630 (25.0%) | **2,515 (100.0%)** |
| **Fixed combination** | 122 (30.3%) | 141 (35.1%) | 139 (34.6%) | **402 (100.0%)** |
| **Free combination** | 121 (23.4%) | 279 (54.1%) | 116 (22.5%) | **516 (100.0%)** |
| **Total** | 871 (25.4%) | 1,677 (48.8%) | 885 (25.8%) | **3,433 (100.0%)** |
| *Total population* | | | | |
| **Monotherapy** | 2,852 (19.1%) | 6,325 (42.3%) | 5,771 (38.6%) | **14,948 (100.0%)** |
| **Fixed combination** | 672 (23.0%) | 984 (33.7%) | 1,263 (43.3%) | **2,919 (100.0%)** |
| **Free combination** | 342 (17.4%) | 1,009 (50.6%) | 644 (32.3%) | **1,995 (100.0%)** |
| **Total** | 3,866 (19.5%) | 8,318 (41.9%) | 7,678 (38.6%) | **19,862 (100.0%)** |

Abbreviations: n, number.

persistence is a more accurate means of characterizing patient commitment and treatment continuity than other indicators usually calculated to describe medication adherence [15]. However, comparison of these findings with those reported for other populations is

**Table 3. Hazard ratios for discontinuation of antihypertensive therapy in new antihypertensive drug users aged ≥40 years in Aragón (Spain).**

| | Unadjusted HR (95%CI) | Adjusted HR (95%CI)** |
|---|---|---|
| *Sex* | | |
| Women | 1 | 1 |
| Men | 1.23 (1.19–1.28)* | 1.20 (1.15–1.24)* |
| *Age group* | | |
| 40–59 | 1 | 1 |
| 60–79 | 1.04 (1.00–1.08) | 1.02 (0.98–1.06) |
| ≥80 | 1.51 (1.44–1.59)* | 1.45 (1.38–1.53)* |
| *Type of therapy* | | |
| Monotherapy | 1 | 1 |
| Fixed combination | 0.79 (0.75–0.84)* | 0.80 (0.76–0.84)* |
| Free combination | 1.01 (0.95–1.08) | 0.99 (0.93–1.06) |

Abbreviations: HR, hazard ratio; 95%CI, 95% confidence interval.

N = 19,543.

*Statistically significant result.

**Model adjusted for all other variables indicated in the table.

complicated by the wide variety of methodologies used to assess persistence [8, 10, 15, 18], including differences in inclusion criteria, wash-out periods, and gap lengths.

Of the new antihypertensive drug users comprising the study population, one fifth exclusively purchased antihypertensive drugs on the index date or during the subsequent 15 days. This observation is in line with the findings of previous studies showing that a considerable number of users filled only one prescription [8]. Patient concerns about the newly prescribed therapy and its potential adverse effects are the most commonly cited reasons for early decrease in persistence [19].

The present study reveals divergences as regards the type of therapy prescribed, with new users receiving monotherapy, fixed combinations, and free combinations accounting for 73.3%, 13.9%, and 9.1% of the study population, respectively. Guidelines published recently by the European Society of Cardiology and the European Society of Hypertension [20] recommend a combination of two drugs with distinct mechanisms of action in high-risk hypertensive patients, as this is related to a faster and more effective blood pressure reduction [21]. This fact could probably explain the higher long-term antihypertensive use in patients starting treatment with a combination of two drugs, compared with initial monotherapy, found in previous studies [22]. Our findings corroborate the protective association between receiving a fixed combination and persistence, compared with monotherapy, but it did not apply to free combinations. Drugs belonging to the agents acting on renin-angiotensin system subgroup were the most prescribed as both monotherapy and fixed combination, while diuretics and beta blockers represented an important proportion of prescriptions within free combinations. Previous studies have consistently described lowest levels of persistence for these latter, and the highest rates for agents acting on the renin-angiotensin system [15, 17, 19, 23]. Additionally, diuretics and beta blockers are often indicated for disorders different than hypertension which, in some cases, require shorter treatment lengths. These situations could explain the lower persistence rates found in the free combination cohort. On the other hand, it seems probable that the different age and clinical profile of free combination users influence on their discontinuation. This hypothesis has been corroborated in the present study since the higher risk of discontinuation in free combination users disappeared after adjusting by sex and age. From a methodological perspective, it should also be taken into account that, differently from the other study cohorts, persistence in the free combination cohort was described as continuation with any of the formulations prescribed, and that free combination cohort was the only one in which the proportion of persistent users varied importantly when a gap of 60 days was applied.

In our study population, the youngest hypertensive drug users (aged 40–59 years) were more often initially treated with fixed combinations, while those aged ≥80 years were more often prescribed free combinations. Multivariate analysis of the effect of age on persistence additionally revealed that the risk of treatment discontinuation was 45% greater in the oldest (≥80 years) than the 40–59 years' group, independently of the type of therapy. While contrasting results have been reported by some authors [8, 10], other studies have described similar age-related effects. For instance, Choi et al. [23] concluded that persistence was greater in elderly patients than young/adult patients, but was poorest in the very elderly. The poorer persistence with increasing age and the greater complexity of these patients may be linked to the presence of other medical conditions that require attention and a great number of comedications. Moreover, control of hypertension may not be considered a priority in these patients, especially in those closer to the end of life.

The existing literature is somewhat ambiguous as regards the influence of sex on antihypertensive drug treatment persistence [8, 10, 12, 23]. In this respect, our results showed that women were less likely than men to discontinue treatment.

Several aspects of our study strengthen the validity of our findings. First, the use of an administrative electronic database allowed assessment of patterns of antihypertensive drug utilization in "real life" conditions in a European population. Second, the inclusion in our analysis of incident, but not prevalent antihypertensive drug users, simplified the evaluation of medication-taking behavior. Finally, in contrast to previous studies [12, 24] our definition of persistence included individuals who continued antihypertensive therapy but whose regimens were switched or added to, thereby avoiding underestimation of true persistence.

A key limitation of our study is the lack of clinical, prescriber or patient socioeconomic information in the data source used, which could have allowed to ascertain the cause of treatment discontinuation in numerous cases, such as the oldest users. Moreover, the availability of information on other pharmacological treatments would have been useful for assessing the effect of polypharmacy on persistence, a relationship with inconsistent findings to date [8, 18]. Pharmacy claim databases like the used do not indicate whether the patient has taken the purchased medication. In this sense, the quantification of persistence seeks to overcome this limitation by providing a measurement of treatment continuity [15]. The use of DDD to estimate the number of days of available medication assumes that all patients are taking the standard dose of a given drug. This can be problematic in cases of non-standard regimens, more common in very elderly patients, or in patients to whom the medication is prescribed for conditions other than hypertension.

To conclude, persistence with antihypertensive drug treatment in our study population was poor overall, and the very elderly users were the less likely to remain persistent. According to our findings, fixed combinations seem an appropriate choice as initial therapy regarding the persistence, and its prescribing is in line with current recommendations. Users of free combinations have shown the lowest frequency of persistence, association probably influenced not only by their different age, but also by their clinical profile and the type of drugs that they receive, compared with the other users. Further research controlling for a higher number of potentially influencing factors than the included in the present study is thus required to confirm this scenario. In any case, the antihypertensive regimen should always be selected based on the patient's circumstances and clinical status, with patients being included in the decision-making process. Finally, patients who are prescribed therapies and regimens that are more likely to be discontinued should be identified and targeted for interventions.

## Supporting information

**S1 Table. Sensitivity analyses: Proportion of persistent antihypertensive users and median time to discontinuation according to the initially prescribed type of therapy when applying gaps of 30, 60 and 90 days.**
(DOCX)

## Acknowledgments

The authors thank O. Howard for English-language editing of the manuscript.

## Author Contributions

**Conceptualization:** Sara Malo, María Jesús Lallana, María José Rabanaque.

**Data curation:** Javier Armesto.

**Formal analysis:** Sara Malo, Cristina Feja.

**Funding acquisition:** María José Rabanaque.

**Methodology:** Sara Malo, Isabel Aguilar-Palacio.

**Supervision:** María José Rabanaque.

**Writing – original draft:** Sara Malo.

**Writing – review & editing:** Isabel Aguilar-Palacio, Cristina Feja, María Jesús Lallana, Javier Armesto, María José Rabanaque.

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
