## [Decision Letter · Decision Letter 0]

19 Aug 2020

PONE-D-20-09984

Effect of patient and treatment factors on persistence with antihypertensive treatment: a population-based study

PLOS ONE

Dear Dr. Malo,

Thank you for submitting your manuscript to PLOS ONE. After careful consideration, we feel that it has merit but does not fully meet PLOS ONE’s publication criteria as it currently stands. Therefore, we invite you to submit a revised version of the manuscript that addresses the points raised during the review process.

We look forward to receiving your revised manuscript.

Kind regards,

Vijayaprakash Suppiah, PhD

Academic Editor

PLOS ONE

Journal Requirements:

Reviewers' comments:

Reviewer's Responses to Questions

**Comments to the Author**

1. Is the manuscript technically sound, and do the data support the conclusions?

Reviewer #1: Partly

Reviewer #2: Yes

2. Has the statistical analysis been performed appropriately and rigorously? 

Reviewer #1: Yes

Reviewer #2: Yes

3. Have the authors made all data underlying the findings in their manuscript fully available?

Reviewer #1: Yes

Reviewer #2: No

4. Is the manuscript presented in an intelligible fashion and written in standard English?

Reviewer #1: Yes

Reviewer #2: Yes

5. Review Comments to the Author

Reviewer #1: The manuscript introduces and important topic relating to persistence among patients using medications for hypertension, a fairly common illness and a risk factor for cardiovascular disease.

My comments are/suggestions are :

1. In the introductions section, the manuscript does not seem to differentiate between the terms adherence or persistence, and reports he rates as though they are one. More details are needed to show the difference and report different rates for each from literature.

2. In the methods section, there is no clear rationale for the definition of the gap utilized, including a reference form previous research that has used this definition would be helpful.

3. Please explain why the authors chose 15 days after index date, most prescriptions in US are for 30, 90 days, if that is not the case in Spain, if so , should be explained

4. The gap lengths used in the study 30,60, 90 days need to be supported by previous literature.

5. The number of variables controlled for in the the multivariate analysis are limited to gender and age. There are so many other factors that influence the adherence of patients as well as persistence that are not included and the uncontrolled confounding can have a significant impact on the study findings. This includes number of other medications, regiment complexity, severity of illnesses, comorbidities, socioeconomic status , geographic location, physician specialty and others. The discussion should be rewritten to highlight this major limitation and perhaps tone down the conclusion to reflect that it was only controlled for a few variables and future studies should ascertain the associations controlling for the lacking variables .

6. In the limitations section, other limitations to list is that refill data is not the same as actual consumption of the medication and the limited generalizability to similar populations

Reviewer #2: The research covers an important topic and has strength that it covers all respective subjects of the condition in the database for a period of time. It is a strength since frequently we see study populations where some sort of convenience sampling is performed. The study covers an important real-life variable - medication presentation form (single, fixed combo or free combo). The methodology is described clearly and in detail.

Study limitations are described - purchase does not necessarily mean correct use and guideline DDDs does not necessarily mean the way it was prescribed by the doctor. However, I would debate that having additional variables (authors mention some of those) from the database would give additional important insight. I would have recommended polypharmacy as a variable for persistence, grouping for co-morbidities and co-treatments, e.g., other cardiovascular conditions, diabetes, depression, neurodegenerative disease. Such analysis would certainly have added to the depth of the study and its conclusions. Such data could have explained the study finding that risk of treatment discontinuation was greater in the oldest.

6. PLOS authors have the option to publish the peer review history of their article (what does this mean?). If published, this will include your full peer review and any attached files.

Reviewer #1: No

Reviewer #2: **Yes: **Dins Smits

---

## [Author Response · Author response to Decision Letter 0]

23 Sep 2020

EFFECT OF PATIENT AND TREATMENT FACTORS ON PERSISTENCE WITH ANTIHYPERTENSIVE TREATMENT: A POPULATION-BASED STUDY (PONE-D-20-09984)

First, we would like to thank the academic editor and the reviewers for the provided comments, which have been very useful to improve this last version of the paper. The modifications made appear highlighted in the marked-up copy of the manuscript and are integrated in the unmarked version. Next, we proceed to answer in detail all the points addressed in the received letter:

Journal requirements

Answer from authors: 

As requested, the journal´s style requirements have been carefully checked and applied to the manuscript.

References have been updated after the application of the modifications requested.

2. We note that you have indicated that data from this study are available upon request. PLOS only allows data to be available upon request if there are legal or ethical restrictions on sharing data publicly. In your revised cover letter, please address the following prompts:

Answer from authors: 

There exist restrictions on publicly sharing the data that we used, so we have provided the following information related. 

In the Methods section, the ethical information has been rewritten as: “The research protocol of the study was approved by the Aragon Health Sciences Institute (IACS) and the Aragon Health Service, who provided us the corresponding pharmacy claims data. All data used were anonymized and the results aggregated, making patient identification impossible. Neither consent to participate nor additional ethics approval were, therefore, needed.”

Regarding the data availability, the permission obtained from the Aragon Health Sciences Institute (IACS) imply the exclusive use of the data by researchers who authored the present study. Thus, this information cannot be published or shared with other institutions. Data access requests should be addressed to the IACS through https://www.iacs.es/.

 Answer from authors: 

In order to fulfil the journal´s data sharing requirements, the phrase “data not shown” has been removed. The results corresponding to the sensitivity analyses have been included into a table as a Supporting information file (S1 Table). Also, a few minor corrections/clarifications have been added to the corresponding paragraph in the Results section:

“In the sensitivity analyses (S1 Table), the proportions of new antihypertensive users considered treatment persistent were 1,7%, 42.3% and 59.0% applying gaps of 30, 60 and 90 days, respectively. These results should be compared with the 38.6% of persistent users obtained when applying a gap of 2xDDD. Median time to discontinuation was of 46 and 228 days with gaps of 30 and 60 days, respectively. For the gap of 90 days and also for the gap of 60 days in the case of the free combination cohort, median time could not be calculated since less than 50% of users had discontinued 365 days after initializing treatment. The proportion of persistent users obtained with the gap of 60 days was similar to the obtained with the gap of 2xDDD for all the types of therapy except for the free combination cohort, where 51.4% of users were classified as persistent, instead of 32.3%.”

Reviewer 1

The manuscript introduces and important topic relating to persistence among patients using medications for hypertension, a fairly common illness and a risk factor for cardiovascular disease.

My comments are/suggestions are:

1. In the introductions section, the manuscript does not seem to differentiate between the terms adherence or persistence, and reports he rates as though they are one. More details are needed to show the difference and report different rates for each from literature.

Answer from authors: 

We agree with the reviewer that the concept of “persistence” should be explained more in depth and differentiated from “adherence”, which represents a broader concept. For this reason, we have included the following explanation in the Introduction section: 

“The European Ascertaining Barriers to Compliance (ABC) project developed a taxonomy to standardize the medication-taking behaviour terminology and the evaluation of medication adherence [3]. According to this taxonomy, adherence is the process by which patients take their medications as prescribed, and has three different components or phases: initiation, implementation and discontinuation. Persistence with therapy refers to the last phase, and it is defined as the time between initiation of therapy and the last dose, which immediately precedes discontinuation. “

We have also modified the following sentences in the Discussion section:

“Evidence suggests that measurement of persistence is a more accurate means of characterizing patient commitment and treatment continuity than other indicators usually calculated to describe medication adherence [15].”

“This fact could probably explain the better long-term adherence to higher long-term antihypertensive use in patients starting treatment with a combination of two drugs, compared with initial monotherapy, found in previous studies [22].”

“Pharmacy claim databases like the used do not indicate whether the patient has taken the purchased medication. In this sense, the quantification of persistence rather than adherence seeks to overcome this limitation by providing a measurement of treatment continuity [15].”

2. In the methods section, there is no clear rationale for the definition of the gap utilized, including a reference form previous research that has used this definition would be helpful.

Answer from authors: 

As requested, we have included a sentence and two references corresponding to studies in which participants were deemed non-persistent if a gap more than twice the duration of the preceding prescription was recorded. The modified sentence is: “Antihypertensive drug users were considered non-persistent or discontinuers if, during the follow-up period, the database revealed a gap more than twice the duration of the preceding prescription during which no medication was dispensed. This gap length was based on previous studies [15-16]. The number of days of available medication was estimated based on the number of DDD in the most recently dispensed prescription. Therefore, the maximum gap allowed was estimated as twice the number of DDD.”

3. Please explain why the authors chose 15 days after index date, most prescriptions in US are for 30, 90 days, if that is not the case in Spain, if so, should be explained.

Answer from authors: 

In our study, each patient was classified into one of the four cohorts (monotherapy, fixed combination, free combination or multiple mix) based on all antihypertensive drugs prescribed to him/her during the 15-day period after the index date, i.e. the index period. This consideration was taken based on the fact that antihypertensive therapy is initiated, in numerous patients, in a gradual manner, depending on the presence/absence of adverse effects, patient needs and response. We believe, therefore, that this consideration reduces the risk of classifying a patient into an erroneous cohort and avoids misclassification of certain persistent users as non-persistent. 

A clarification of this matter has been added in the Outcome variables subsection: 

“The consideration of the index period was based on the fact that antihypertensive therapy prescribing is initiated in a gradual manner in certain patients, depending on their needs and response. Taking into account all drugs prescribed during this first period may, therefore, reduce the risk of misclassification of antihypertensive drug users.”

4. The gap lengths used in the study 30, 60, 90 days need to be supported by previous literature.

Answer from authors: 

Although our study analyses were performed by considering a gap length defined according to the number of doses supplied, rather than a fixed number of days, we also conducted a sensitivity analysis to prove the robustness of the method. In this analysis, we replicated the previous persistence analyses by applying gap lengths of 30, 60 and 90 days. These lengths were based on the used in previous similar studies, whose references have been incorporated to the Statistical analysis subsection:

“A sensitivity analysis was performed to account for the possible influence of the selected gap length and the number of DDD as a measure of the daily dose on participant classification. The proportion of persistent users and the median time to discontinuation were recalculated for each cohort by applying fixed gap lengths of 30, 60 and 90 days, based on previous research [9, 16-17].”

5. The number of variables controlled for in the multivariate analysis are limited to gender and age. There are so many other factors that influence the adherence of patients as well as persistence that are not included and the uncontrolled confounding can have a significant impact on the study findings. This includes number of other medications, regiment complexity, severity of illnesses, comorbidities, socioeconomic status, geographic location, physician specialty and others. The discussion should be rewritten to highlight this major limitation and perhaps tone down the conclusion to reflect that it was only controlled for a few variables and future studies should ascertain the associations controlling for the lacking variables.

Answer from authors: 

We coincide with the reviewer that the availability and analyses of the mentioned factors which are potentially associated with persistence would have probably gave rise to a clearer and more valid interpretation of the study findings. Their lack represents a major limitation, therefore the following sentence has been modified as follows: “A key limitation of our study is the lack of clinical information in the data source used, which could have allowed to ascertain the cause of treatment discontinuation in numerous cases, such as the oldest users. Moreover, the availability of information on the physician specialty, the patient socioeconomic level or other pharmacological treatments would have been useful. In the latter case, for assessing the effect of polypharmacy on persistence, a relationship with inconsistent findings to date [8, 18]. 

Additionally, the last paragraph in the Discussion has been changed into: “Users of free combinations have shown the lowest frequency of persistence, association probably influenced not only by their different age, but also by their clinical profile and the type of drugs that they receive, compared with the other users. Further research controlling for a higher number of potentially influencing factors than the included in the present study is thus required to confirm this scenario.”

6. In the limitations section, other limitations to list is that refill data is not the same as actual consumption of the medication and the limited generalizability to similar populations

Answer from authors: 

In the limitations section, we acknowledge that purchase does not mean consumption: “Pharmacy claim databases like the used do not indicate whether the patient has taken the purchased medication. In this sense, the quantification of persistence seeks to overcome this limitation by providing a measurement of treatment continuity [15].”

To our knowledge, most of the observational studies aimed at assessing medication taking behavior in patients with chronic diseases are performed by using pharmacy claims data, which are particularly useful for the evaluation of drugs intended for long term therapy such as antihypertensive drugs. For this reason, we do not consider that the use of our data source jeopardize the comparison/generalizability of the presented results. If the reviewer disagrees with this reasoning or considers that a related comment should be included in the manuscript, we are open to suggestions.

Reviewer 2

The research covers an important topic and has strength that it covers all respective subjects of the condition in the database for a period of time. It is a strength since frequently we see study populations where some sort of convenience sampling is performed. The study covers an important real-life variable - medication presentation form (single, fixed combo or free combo). The methodology is described clearly and in detail.

Study limitations are described - purchase does not necessarily mean correct use and guideline DDDs does not necessarily mean the way it was prescribed by the doctor. However, I would debate that having additional variables (authors mention some of those) from the database would give additional important insight. I would have recommended polypharmacy as a variable for persistence, grouping for co-morbidities and co-treatments, e.g., other cardiovascular conditions, diabetes, depression, neurodegenerative disease. Such analysis would certainly have added to the depth of the study and its conclusions. Such data could have explained the study finding that risk of treatment discontinuation was greater in the oldest.

Answer from authors: 

We agree with the reviewer that having additional information would have probably contributed to a clearer and more valid interpretation of the study findings, especially in cases such as the oldest users. Thus, we have modified the following sentence as follows: “A key limitation of our study is the lack of clinical information in the data source used, which could have allowed to ascertain the cause of treatment discontinuation in numerous cases, such as the oldest users. Moreover, the availability of information on the physician specialty, the patient socioeconomic level or other pharmacological treatments would have been useful. In the latter case, for assessing the effect of polypharmacy on persistence, a relationship with inconsistent findings to date [8, 18]. 

Additionally, the last paragraph in the Discussion has been modified: “Users of free combinations have shown the lowest frequency of persistence, association probably influenced not only by their different age, but also by their clinical profile and the type of drugs that they receive, compared with the other users. Further research controlling for a higher number of potentially influencing factors than the included in the present study is thus required to confirm this scenario.”

---

## [Decision Letter · Decision Letter 1]

23 Dec 2020

PONE-D-20-09984R1

Effect of patient and treatment factors on persistence with antihypertensive treatment: a population-based study

PLOS ONE

Dear Dr. Malo,

Thank you for submitting your manuscript to PLOS ONE. After careful consideration, we feel that it has merit but does not fully meet PLOS ONE’s publication criteria as it currently stands. Therefore, we invite you to submit a revised version of the manuscript that addresses the points raised during the review process.

The points raised by the reviewers are adequately addressed. However, the following sentence does not seem to be correct: "In the latter case, for assessing the effect of polypharmacy on persistence, a relationship with inconsistent findings to date [8, 18]." As the is no copy-editing provided by PLOS One, the text must be copy-edited by the authors. I would like to ask you to correct this sentence and to very carefully re-read your article and copy-edit it. Thank you.

We look forward to receiving your revised manuscript.

Kind regards,

Hans-Peter Brunner-La Rocca, M.D.

Academic Editor

PLOS ONE

Reviewers' comments:

Reviewer's Responses to Questions

**Comments to the Author**

1. If the authors have adequately addressed your comments raised in a previous round of review and you feel that this manuscript is now acceptable for publication, you may indicate that here to bypass the “Comments to the Author” section, enter your conflict of interest statement in the “Confidential to Editor” section, and submit your "Accept" recommendation.

Reviewer #2: All comments have been addressed

2. Is the manuscript technically sound, and do the data support the conclusions?

Reviewer #2: Yes

3. Has the statistical analysis been performed appropriately and rigorously? 

Reviewer #2: Yes

4. Have the authors made all data underlying the findings in their manuscript fully available?

Reviewer #2: No

5. Is the manuscript presented in an intelligible fashion and written in standard English?

Reviewer #2: Yes

6. Review Comments to the Author

Reviewer #2: (No Response)

7. PLOS authors have the option to publish the peer review history of their article (what does this mean?). If published, this will include your full peer review and any attached files.

Reviewer #2: **Yes: **Dins Smits

---

## [Author Response · Author response to Decision Letter 1]

24 Dec 2020

EFFECT OF PATIENT AND TREATMENT FACTORS ON PERSISTENCE WITH ANTIHYPERTENSIVE TREATMENT: A POPULATION-BASED STUDY (PONE-D-20-09984)

First, we would like to thank the academic editor and the reviewers for the provided comments, which have been very useful to improve the last version of the paper. The modifications made appear highlighted in the marked-up copy of the manuscript and are integrated in the unmarked version. Next, we explain the modifications applied based on the received letter:

The points raised by the reviewers are adequately addressed. However, the following sentence does not seem to be correct: "In the latter case, for assessing the effect of polypharmacy on persistence, a relationship with inconsistent findings to date [8, 18]." As the is no copy-editing provided by PLOS One, the text must be copy-edited by the authors. I would like to ask you to correct this sentence and to very carefully re-read your article and copy-edit it. Thank you.

Answer from authors: 

Thank you for the comment. As requested, the sentence has been corrected: “Moreover, the availability of information on other pharmacological treatments would have been useful for assessing the effect of polypharmacy on persistence, a relationship with inconsistent findings to date [8, 18]. “

Moreover, the manuscript has been carefully re-read and some modifications have been introduced in order to improve its writing, wording and understanding.

---

## [Editor Report · Decision Letter 2]

5 Jan 2021

Effect of patient and treatment factors on persistence with antihypertensive treatment: a population-based study

PONE-D-20-09984R2

Dear Dr. Malo,

We’re pleased to inform you that your manuscript has been judged scientifically suitable for publication and will be formally accepted for publication once it meets all outstanding technical requirements.

Kind regards,

Hans-Peter Brunner-La Rocca, M.D.

Academic Editor

PLOS ONE
---

## [Editor Report · Acceptance letter]

7 Jan 2021

PONE-D-20-09984R2 

Effect of patient and treatment factors on persistence with antihypertensive treatment: a population-based study 

Dear Dr. Malo:

I'm pleased to inform you that your manuscript has been deemed suitable for publication in PLOS ONE. Congratulations! Your manuscript is now with our production department. 

Kind regards, 

on behalf of

Dr. Hans-Peter Brunner-La Rocca 

Academic Editor

PLOS ONE